# Radar Characteristics and Causal Analysis of Two Consecutive Tornado Events Associated with Heavy Precipitation during the Mei-Yu Season

**Shuya Cao** [1,2,3], **Yi Wang** [2,4], **Guangxin He** [3,5,*], **Peifeng Shen** [1,2], **Yan He** [1,2] and **Yue Wu** [1]

1    Suzhou Meteorological Bureau, Suzhou 215131, China; caoshuya0929@163.com (S.C.);
     peifeng_shen78@outlook.com (P.S.); mhymcathy@gmail.com (Y.H.); orangewrq@sina.com (Y.W.)
2    Key Laboratory of Transportation of Meteorology CMA, Nanjing 210019, China; wangyi401@163.com
3    Guangzhou Institute of Tropical Marine Meteorology, China Meteorological Administration,
     Guangzhou 510080, China
4    Jiangsu Meteorological Observatory, Nanjing 210019, China
5    Key Laboratory of Meteorological Disaster, Ministry of Education (KLME), International Joint Research
     Laboratory on Climate and Environment Change (ILCEC), Collaborative Innovation Center on Forecast and
     Evaluation of Meteorological Disasters, Nanjing University of Information Science and Technology,
     Nanjing 210044, China
*    Correspondence: 002421@nuist.edu.cn; Tel.: +86-1339-0754-007

**Abstract:** This paper comprehensively analyzed two consecutive tornado events associated with heavy precipitation during the Mei-yu season (a period of continuous rainy weather that occurs in the middle and lower reaches of the Yangtze River in China from mid-June to mid-July each year) and detailed the formation and development process of the tornadoes using Doppler weather radar, wind profiler radar, ERA5 reanalysis data, ground automatic station data and other multi-source data. The results showed that: (1) Small-scale vortices were triggered and developed during the eastward movement of the low vortex, forming two tornadoes successively on the eastern section of the Mei-yu front. (2) The presence of a gap on the front side of the reflectivity factor profile indicated that strong incoming airflow entered the updraft. Mesocyclones were detected with decreasing heights and increasing shear strengths. The bottom height of the tornado vortex signature (TVS) dropped to 0.7 km, and the shear value increased to $55.4 \times 10^{-3}$ s$^{-1}$. Tornado debris signatures (TDSs) could be seen with a low cross-correlation coefficient (CC) value area of 0.85–0.9 in the mesocyclone. The difference between the lowest-level difference velocity (LLDV) and the maximum difference velocity (MXDV) reached the largest value when a tornado occurred. (3) The continuously enhanced low-level jet propagated downward to form a super-low-level jet, and the strong wind direction and wind speed convergence in the boundary layer created a warm, moist and unstable atmosphere in Suzhou. With the entrainment of dry air, the northwest dry jet and the southeast moist jet stimulated the formation of a miniature supercell. (4) The low-level vertical wind shear of 0–1 km increased significantly upon tornado occurrence, which was more conducive to the formation and intensification of horizontal vorticity tubes. Encountering updrafts and downdrafts, the vorticity tubes might have been stretched and intensified. The first lightning jumps appeared 15 min and 66 min earlier than the Kunshan Bacheng tornado and the Taicang Liuhe tornado. The Liuhe tornado occurred during the stage when the lightning frequency reached its peak and then fell back.

**Keywords:** radar observations; tornado; heavy rainfall supercell storm; TVS; dual polarization Doppler weather radar; LLDV

## 1. Introduction

Tornadoes are the most intense vortex phenomena in the atmosphere, generally manifesting as high-speed rotating "funnel" clouds extending from a thunderstorm cloud to the underlying surface. They are often triggered by strong convective storms, characterized by

small scale, rapid generation and dissipation, and severe disaster. Supercell tornadoes are usually produced by supercell storms and are associated with mesocyclones, and persistent mesocyclones are easily induced by tornadoes in environments with large wind shears and low lifting condensation heights in the 0–1 km range [1]. Strong tornadoes are often associated with mesocyclones and TVSs, which are smaller in scale than mesocyclones, and the probability of tornado occurrence is significantly increased when mesocyclones and TVSs are detected simultaneously [2]. Many meteorologists have analyzed high-temporal and -spatial-resolution weather radar observation data and summarized many radar reflectivity characteristics of supercells: hook echoes, bounded weak echo regions and pendant echo structures [3–7]. In the case of X-band polarimetric radar detection of tornadoes, the correlation coefficients of non-meteorological echoes (annular tornado debris zones) are significantly smaller than the observed values of meteorological echoes [8]. The magnitude of the specific differential phase (abbreviated as $K_{DP}$) mainly depends on the content of liquid water condensate, which is very useful for locating strong precipitation areas and quantitative precipitation estimation [9].

Tornadoes in China often occur under weather backgrounds such as the peripheries of typhoons [10], upper-level cold vortices [11], and Mei-yu fronts [12], and some strong tornadoes often occur in environments with high convective available potential energy (CAPE) and strong low-level vertical wind shear [13], similar to those in foreign countries [14–16]. The Jiangsu–Anhui Plain is one of the areas with the most frequent tornado occurrence in China, with most occurrences in June-August [17]. In early July, the Jianghuai region is in the Mei-yu period, with frequent heavy rainfall, accompanied by tornadoes that occur in heavy rainfall during the Mei-yu period, accounting for about 30% of the tornadoes in Jiangsu [12]. More than 50% of the tornadoes in Jiangsu are produced within the mesocyclones of supercells embedded in multi-cell storm systems, and 30% of the tornadoes are produced in quasi-linear convective systems [18]. China has gained a deeper understanding of the small-scale characteristics of supercell storms that generate tornadoes, such as relatively weak surface cold pools, mesocyclone base heights typically below 1 km, positive correlation between intensity and tornado intensity, tilt, tornado debris characteristics, and sink reflectivity factor cores, and the multi-vortex characteristics of some tornadoes in recent years [19]. Mesocyclones lasting for three volume scans and mesocyclones with low centroids and strong shears are conducive to the production of tornadoes [20]. When the minimum elevation velocity difference of a TVS exceeds 23 m·s$^{-1}$, it is more likely to generate strong tornadoes [21]. Lightning activity could be an indicator of the occurrence of a tornado [22].

Although the above research results have improved understanding of the tornado weather generation environment and the internal wind field of storms to a certain extent, the reasons for the occurrence of tornado weather, especially that accompanied by heavy rainfall during the Mei-yu period, are very complicated and not well studied yet. In addition, there are few studies on tornado cases that occur in southern Jiangsu; therefore, this paper selects two consecutive tornado events associated with heavy precipitation during the Mei-yu season in Suzhou for study. Based on multi-source observation data, such as radar, wind profile, microwave radiometer, ground automatic station, and hourly reanalysis data of the European Numerical Forecast Center ERA5, the environmental conditions that triggered tornadoes, the occurrence and development of convective systems, the evolution characteristics of mesocyclones and TVSs, etc., are analyzed in detail in order to provide a reference for the monitoring and early warning of such Mei-yu-period tornadoes in southern Jiangsu and to reduce tornado disaster losses from a meteorological perspective. However, due to the insufficient monitoring density and the limitations of detection technology, the study of tornadoes is still difficult in mesoscale meteorology, and the following analysis is only a preliminary discussion.

## 2. Materials and Methods

### 2.1. Materials

The data used in this paper include: (1) the fifth-generation global atmospheric reanalysis data ERA5 (ECMWF Reanalysis v5) released by the European Centre for Medium-Range Weather Forecasts (ECMWF), with a spatial resolution of 0.1 × 0.1° and a temporal resolution of 1 h "URL (accessed on 15 May 2022), https://confluence.ecmwf.int/display/CKB/ERA5"; (2) the minute precipitation, temperature, and 2 min mean wind direction and speed of the automatic stations within Jiangsu Province (Figure 1) from 00:00 to 12:00 on 6 July 2020; (3) the Doppler weather radar data of Changzhou (CZ) and Nantong (NT) from 00:00 to 12:00 on 6 July 2020; (4) the wind profile data of Taicang (TC) from 00:00 to 12:00 on 6 July 2020; and (5) the microwave radiometer data of Kunshan (KS) from 00:00 to 12:00 on 6 July 2020.

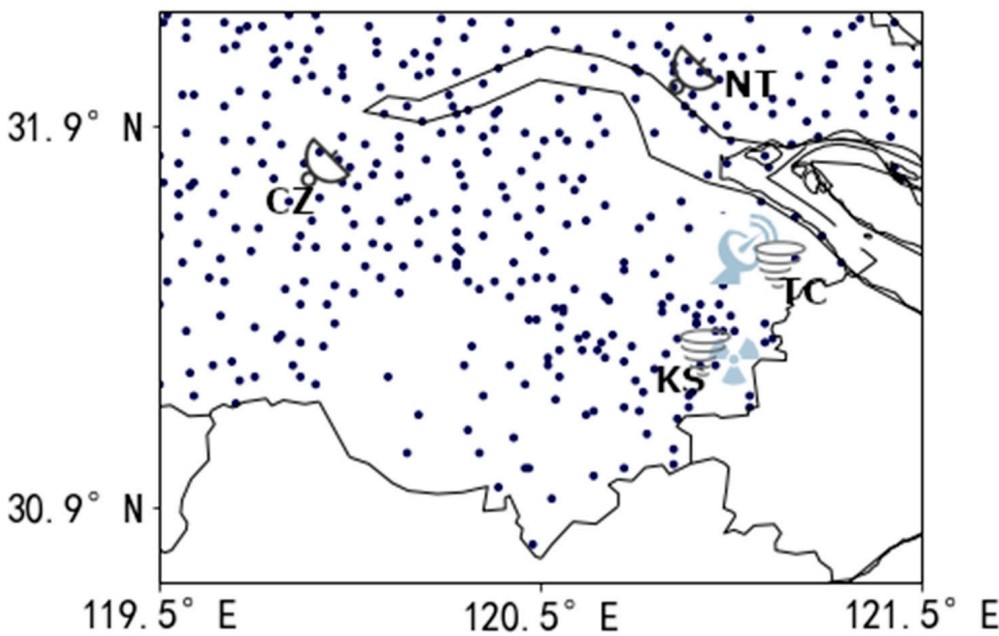

**Figure 1.** Distribution of meteorological stations (black dots) in Jiangsu Province and the location of the Changzhou Doppler radar (CZ), the Nantong Doppler radar (NT), the Taicang wind profile radar (TC) and the Kunshan microwave radiometer (KS).

### 2.2. Methods

In this paper, conventional weather charts and ERA5 reanalysis data are used to analyze the synoptic-scale circulation background and explore the large-scale features of heavy rain and tornado occurrence and development; the characteristics of the mesoscale low vortex rainband are revealed by the analysis of the ground automatic stations; and the evolution features of the convective system are further confirmed by cloud images and radar data. Finally, the distribution of thermodynamic and dynamic field physical quantities and the evolution of the low-level jet near the tornado occurrence site are calculated using the ERA5 reanalysis data and the wind profile radar data to reveal the environmental characteristics of the convective development and evolution.

## 3. Results

### 3.1. Overview of the Process and Weather Background

#### 3.1.1. Precipitation Observation

Affected by the quasi-stationary Mei-yu front and the development of the mesoscale low vortex on the Mei-yu front, a rainstorm/heavy rainstorm occurred in Suzhou from 20:00 on 5 July to 20:00 on 6 July 2020 (Beijing time, the same below), with the accumulated rainfall at 89 automatic meteorological stations out of 121 reaching rainstorm to heavy

rainstorm levels, and a maximum daily precipitation of 202.6 mm occurred in Huangdai Town, Xiangcheng District (Figure 2a). Affected by a strong convective cloud cluster, short-term heavy precipitation occurred in the central and southern parts of Suzhou from the early morning to the morning of 6 July, with the hourly rainfall exceeding 20 mm at 102 automatic meteorological stations out of 121, accounting for 84% of the stations, with the hourly rainfall exceeding 50 mm at 16 stations. The precipitation intensity increased rapidly in a short time, with a maximum hourly rainfall of 41.3 mm at Suzhou Station from 08:00 to 09:00 and a maximum hourly rainfall of 72.5 mm at Jiangxiang Village Station in Kunshan from 09:00 to 10:00, breaking the record for hourly rainfall at this station since it was established in 2012 (58.9 mm in 2021) (Figure 2b). This process was characterized by high rainfall efficiency and was of an extreme nature, causing serious waterlogging in some areas of Suzhou.

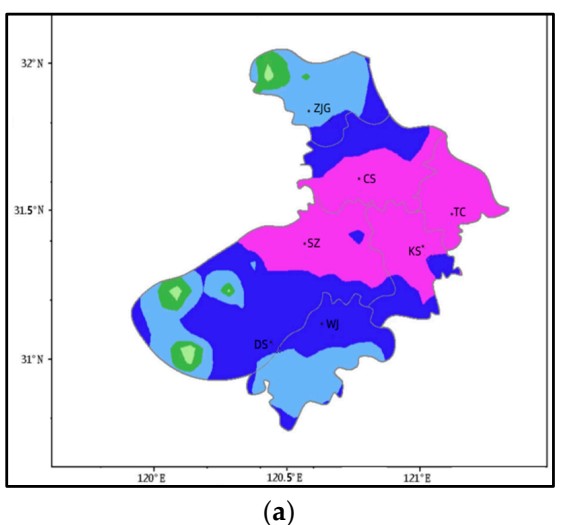

(**a**)

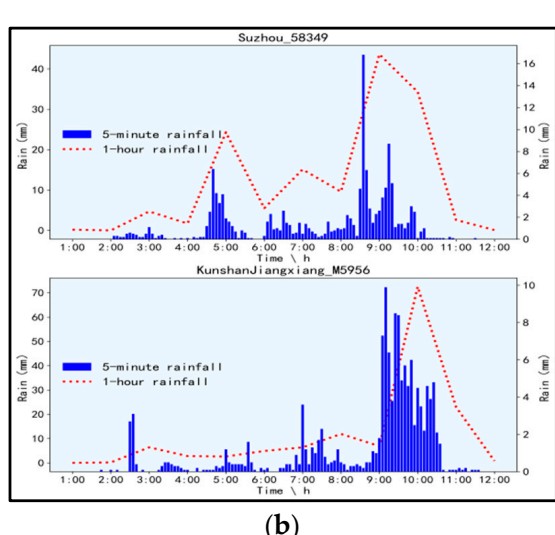

(**b**)

**Figure 2.** Precipitation distribution for Suzhou from 20:00 5 July to 20:00 6 July BT (**a**). Precipitation per 5 min (blue bar) and per hour (red dashed line) for Suzhou and Kunshan Jiangxiang Station from 01:00 to 12:00 BT on 6 July 2020 (**b**).

3.1.2. Tornado Intensity and Path Information

The rainfall in Kunshan increased and strong winds occurred around 9:00 on 6 July 2020. The maximum wind speed of 15.4 m·s$^{-1}$ was detected at Kunshan Station, and the maximum wind speed of 21.7 m·s$^{-1}$ was detected at the station on Yangcheng Lake. Affected by the EF0 tornado (the maximum wind speed of level 9 was detected at the nearby station on Yangcheng Lake) around 9:00 in Chuodunshan Village, Bacheng Town, Kunshan City, 33 households suffered from the disaster, of which 13 households suffered serious damage (Figure 3a). A strong gusting wind of 21.5 m·s$^{-1}$ (level 9) at the automatic station in Liuhe Town, Taicang, Jiangsu, was detected at 9:53, and houses, farmland, trees, etc., were severely damaged after the strong wind, causing serious economic losses but no casualties (Figure 3b). According to the comprehensive analysis of the disaster investigation results and the weather situation and radar data by Professor Zheng Yongguang of the National Meteorological Center and the forecast experts of the Jiangsu Meteorological Bureau, it was determined that a level 3 (industry standard)/(EF2-EF3) intensity tornado with a tornado ground width of about 100 m and a ground path length of about 1 km hit Liuhe Town, Taicang, between 09:54 and 10:00 on 6 July, jumping forward from west to east (Figure 3c).

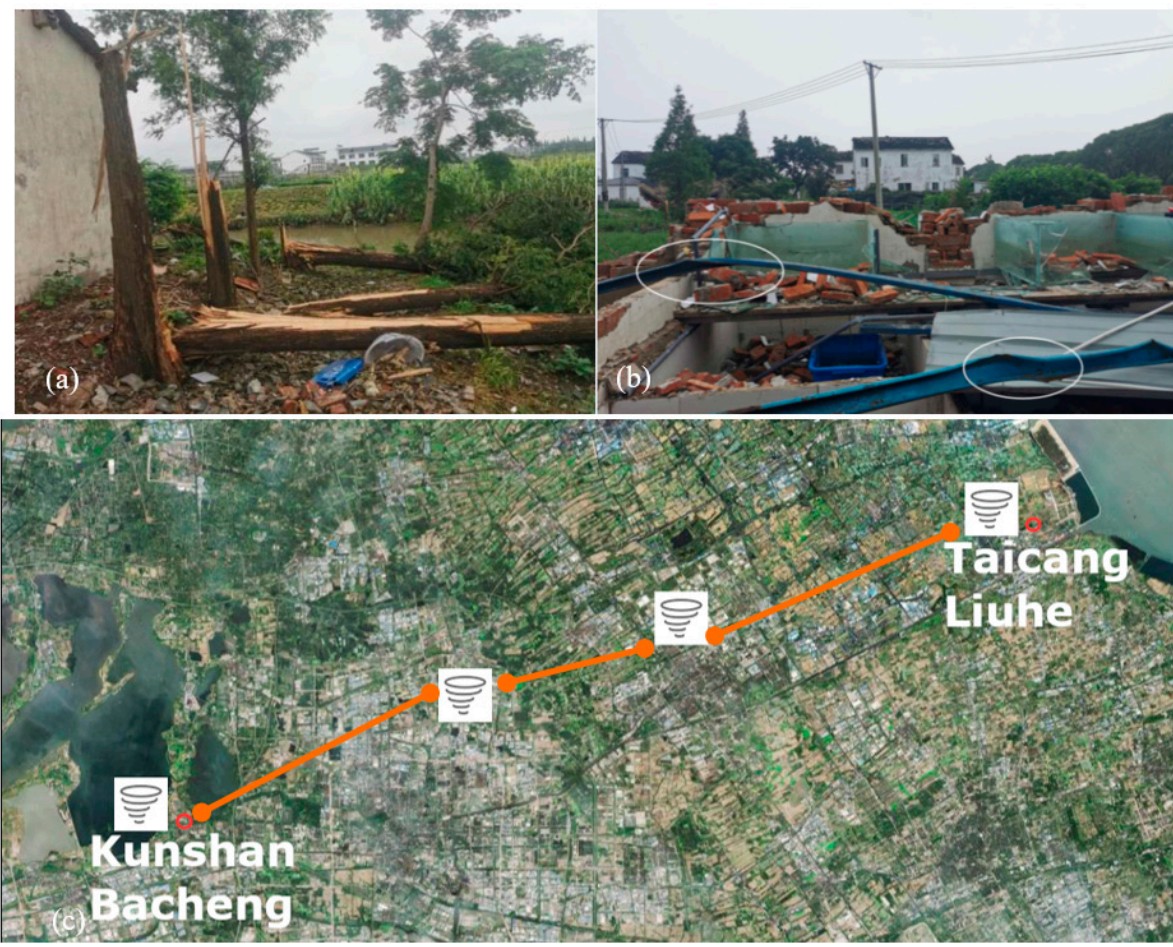

**Figure 3.** Tornado disaster situation (**a**,**b**) of Jiangsu Suzhou on 6 July 2020 and tornado movement path (**c**).

*3.2. Evolution of Convective System*

3.2.1. Initiation Stage of Convective Storm

The evolution of the convective system associated with the tornado and heavy rainfall event can be divided into three stages: the initiation stage of the convective storm (02:00–05:00 BT), the development stage of the convective storm (05:00–08:00 BT) and the mature stage of the supercell storm (08:00–10:00 BT). Precipitation echoes appeared near Xiangcheng (XC) in Suzhou (SZ) at 02:00 BT on 6 June, and the initial echoes remained quasi-stationary near Xiangcheng from 02:00 to 03:00 BT, causing persistent rainfall. At this stage, the rainfall intensity was not high, and the hourly rainfall was less than 10 mm, mainly consisting of stratiform precipitation (Figure 4a). After 03:00 BT, new convective cells were continuously triggered near the echo, and the convective cells showed backward propagation when moving downstream, forming stratocumulus mixed blocky echoes. The echoes continued to develop and gradually evolved into a vortex-shaped convective cloud cluster upstream of Suzhou from 05:00 to 08:00 BT (Figure 4b–d).

3.2.2. Development Stage of the Convective Storm

The vortex-shaped convective cloud cluster developed, the range of the strong echo center in the cloud cluster increased and the center intensity exceeded 55 dBz from 08:24 to 08:30 BT. The stratiform echoes near the rear of the cloud cluster in Wuxi gradually organized into a band-shaped structure. A mesocyclone appeared at a height of 2.4 km, developing upward and then downward, and the convective cells gradually transformed into a supercell later. The broad vortex-shaped echo indicated that the strong rainfall

wrapped the mesocyclones (Figure 5a–c). At 08:42 BT, the vortex-shaped convective cloud cluster continued to intensify and move eastward, with the intensity at the center exceeding 60 dBz (Figure 5d) and the top height exceeding 14 km (Figure 5h). A clear positive–negative velocity couplet appeared on the radial velocity map at the 0.5° elevation angle, with a rotational velocity of 15.8 m·s$^{-1}$ (Figure 5g), corresponding to short-term extreme heavy rainfall of 17.1 mm in 5 min at Suzhou station. A convective cell was triggered on the band-shaped echo originally near Wuxi, and the band-shaped echo structure gradually became compact. The echo top height exceeded 15 km, and the strongest intensity at the center remained above 60 dBz at 08:48 BT (Figure 5e). The strong reflectivity factor area corresponding to the heavy rainfall occurrence coincided with the high K$_{DP}$ area (Figure 5f,i), indicating a high liquid water content. The wind field structure of the vertical vortex caused the precipitation particles to grow rapidly in the continuous up and down tumbling, resulting in the maximum hourly rainfall intensity at Suzhou station from 08:00 to 09:00 BT. The positive–negative velocity couplet persisted, but the rotational velocity slightly decreased.

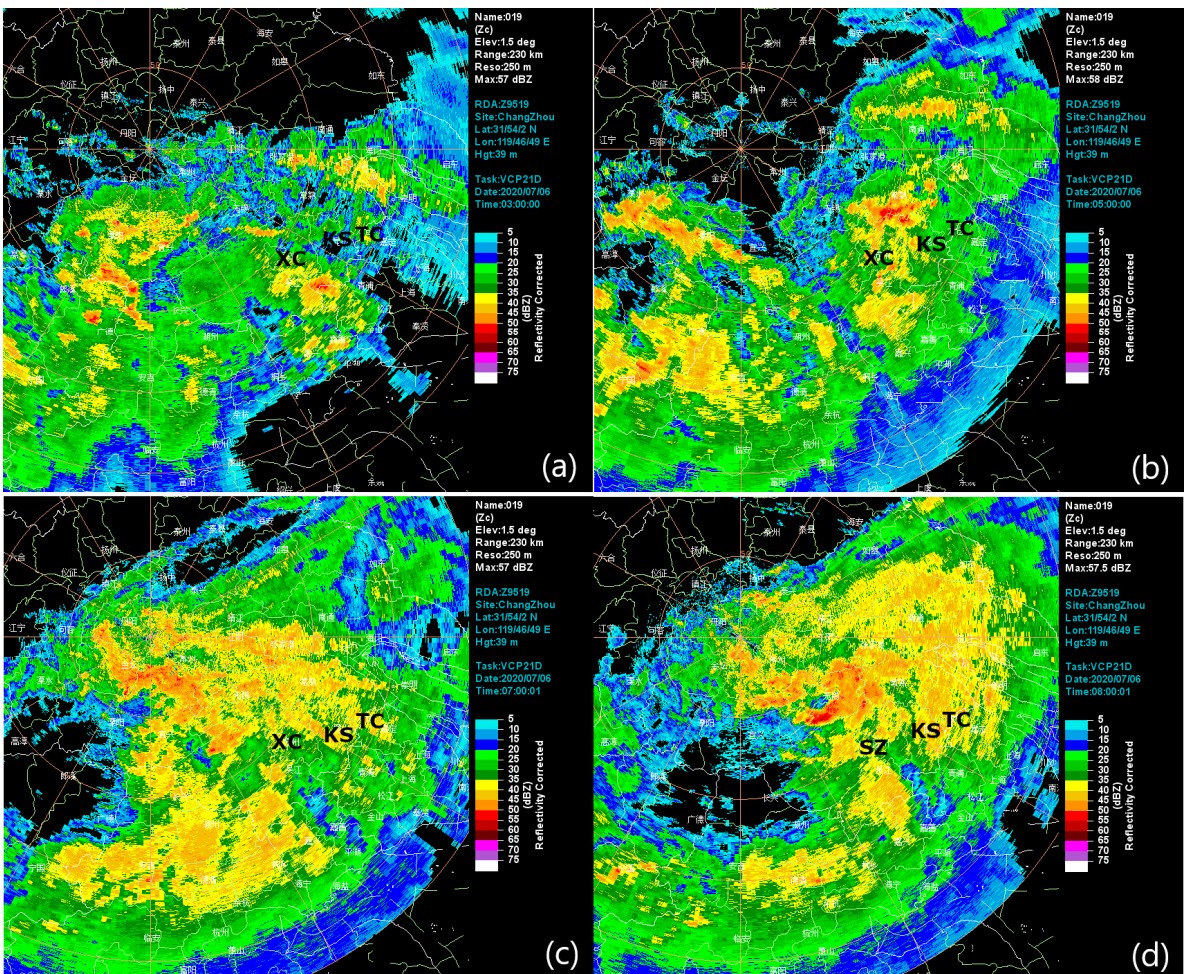

**Figure 4.** The 0.5° elevation basic reflectivity at 3:00 BT (**a**), 5:00 BT (**b**), 7:00 BT (**c**), and 8:00 (**d**) BT on 6 July 2020(SZ represents Suzhou, KS represents Kunshan, TC represents Taicang).

### 3.2.3. Mature Stage of the Supercell

The north–south-oriented echo band moved eastward from the Yangcheng Lake area of Kunshan from 8:54 to 9:00, with the central intensity exceeding 55 dBz, corresponding to the occurrence of short-term heavy precipitation near Taicang in the eastern part of Suzhou. The intensity of the east–west-oriented echo band increased when moving eastward, and new convective cells were continuously generated at the tail. A mesocyclone with a

diameter of 6–8 km appeared at the lowest elevation angle, and the rear flank downdraft (RFD) developed continuously behind (Figure 6a,b). The supercell continued to intensify and evolved into a bow echo with mid-altitude radial convergence (MARC) (Figure 6g,h). The tail convective cells began to merge, and the intensity increased rapidly at 09:06. There was a clear radial convergence zone on the radial velocity map with a vortex feature on the basic reflectivity factor graph at the 0.5° elevation angle, indicating a strong inflow area at the lower level of the storm (Figure 6c). The vertical cross-section showed that the convective echoes exceeding 40 dBz were all below 6 km in height, and the strong echo centroid was low (Figure 6n). The strong echo core shifted to the right with the increase in height. There was a gap in the front, indicating that the strong inflow air entered the updraft. The mesocyclone and TVS products identified the mesocyclone and the TVS, which caused a tornado near Bacheng, Kunshan, Jiangsu Province. The small-scale rotation feature was not obvious, indicating that the supercell was weak (Figure 6c,i).

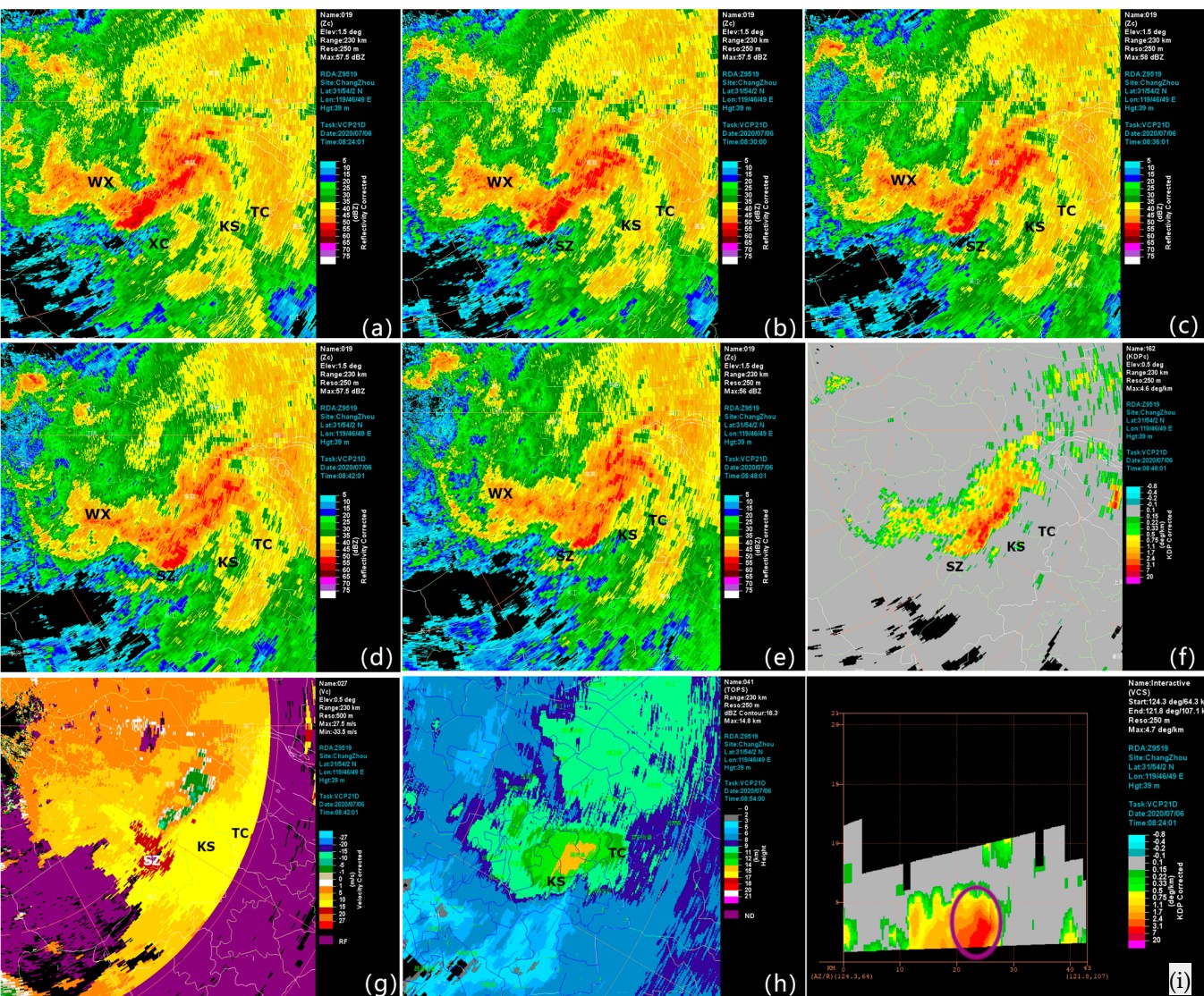

**Figure 5.** The 0.5° elevation basic reflectivity factor at 8:24 BT (**a**), 8:30 BT (**b**), 8:36 BT (**c**), 8:42 BT (**d**), and 8:48 BT (**e**) on 6 July 2020. The 0.5°elevation Kdp at 8:48 (**f**). The 0.5° elevation radial velocity at 8:42 (**g**). The echo top height at 8:48 (**h**). Kdp profile at 08:24 T (The purple elliptical box represents Kdp columns) (**i**). (WX represents Wuxi, SZ represents Suzhou, XC represents Xiangcheng, KS represents Kunshan, TC represents Taicang).

At 09:12, the east–west-oriented band echo began to show a train effect and continued to propagate eastward, with the intensity maintained, resulting in short-term heavy precipitation exceeding 30 mm·h$^{-1}$ in Suzhou (SZ) from 09:00 to 10:00. The mesoscale vortex at the northern end of the bow echo developed continuously with the further intrusion of the dry and cold air from the northwest, and the vortex-shaped echo began to show a hollow structure. The diameter of the original mesocyclone decreased rapidly from 6–8 km to 3–4 km before and after 09:36. The bow echo split completely into a block-shaped echo on the north side and a band-shaped echo on the south side during its eastward movement from 09:30 to 09:36. The band-shaped echo extended southward and moved eastward mainly (Figure 6d). The main body of the echo weakened at 09:42, and the movement direction gradually changed from eastward to southeastward (Figure 6e). A tornado vortex signature (TVS) appeared on the radial velocity map around 09:48, causing a tornado near Liuhe (LH), Taicang (TC), Jiangsu Province. It can be seen from the dual-polarization radar parameter CC that the tornado had a tornado debris signature (TDS) at 09:48. A low CC value area of 0.85–0.9 in the mesocyclones and the abnormally low CC value area can be seen from the CC profile, mainly due to the fact that tornadoes near the ground threw debris into the air in random directions that was irregular in shape, resulting in a small correlation coefficient CC at the TDS feature (Figure 6m). The strong echo at the front of the band-shaped echo moved into Shanghai (SH) at 09:54. The echo intensity in Suzhou decreased significantly, and the organizational structure began to loosen. The precipitation intensity in the western part of Suzhou (SZ) decreased rapidly after 10:00.

Figure 6p provides a schematic diagram of the conceptual model of the Taicang Liuhe tornado during this process. The tornado mainly occurred in the heavy precipitation supercell, with obvious hook-like structural characteristics. The tornado occurred near Liuhe, accompanied by a mesocyclone and a TVS. TDS characteristics appeared near the tornado location, and the high K$_{DP}$ values associated with heavy precipitation were located near the tornado and the inflow area of the southern part of the tornado. Overall, the structure conforms to the classic characteristics of the tornado supercell structure.

### 3.2.4. Evolution of the Mesocyclone and the TVS

Figure 7 shows the evolution trend of the mesocyclone (Figure 7a) and the TVS (Figure 7b) detected by the Nantong Radar Station, in which the x-coordinate represents time, the vertical coordinate on the left represents the height and the shear value is on the right. It can be seen that from 8:40 to 10:06, the mesocyclone and the accompanying TVS features were sustained for five and eight volume scans, respectively, and the possibility of tornado occurrence increased gradually. The base height of the mesocyclone was basically below 1 km 25 min before the tornado occurred near Bacheng, Kunshan. The maximum shear intensity of the TVS reached $55.4 \times 10^{-3} \text{s}^{-1}$ at 8:54, and the shear value continued to decline for three consecutive volume scans. When the tornado occurred at 9:06, the base height of the mesocyclone dropped to 800 m, and the base height of the TVS was 700 m. The height of the maximum shear dropped rapidly from 1.8 km to 700 m, indicating that the tornado vortex was continuously lowering, and the shear intensity was $35.8 \times 10^{-3} \text{ s}^{-1}$.

The base height of the mesocyclone remained below 1.5 km after 09:12, and the shear intensity dropped rapidly. The base height of the TVS remained at 600–700 m, and the top height continued to drop. The height of the maximum shear remained near the base height of the TVS, and the maximum shear value reached $54.3 \times 10^{-3} \text{ s}^{-1}$. From 09:30 to 09:36, the base height of the mesocyclone dropped again and the TVS was not detected during this period. The lowest base height of the mesocyclone was 500 m at 09:36. The hook echo structure of the tornado was clearer at 09:42 before the tornado occurred near Liuhe, Taicang. At the same time, the Nantong radar detected the TVS again, with the shear intensity rising to $50.3 \times 10^{-3} \text{ s}^{-1}$. The height of the maximum shear remained near the base height of the TVS, indicating that the tornado vortex had touched the ground. The Liuhe tornado occurred on the supercell with a mesocyclone on the convergence line in the warm sector at the right front side of the mesoscale vortex.

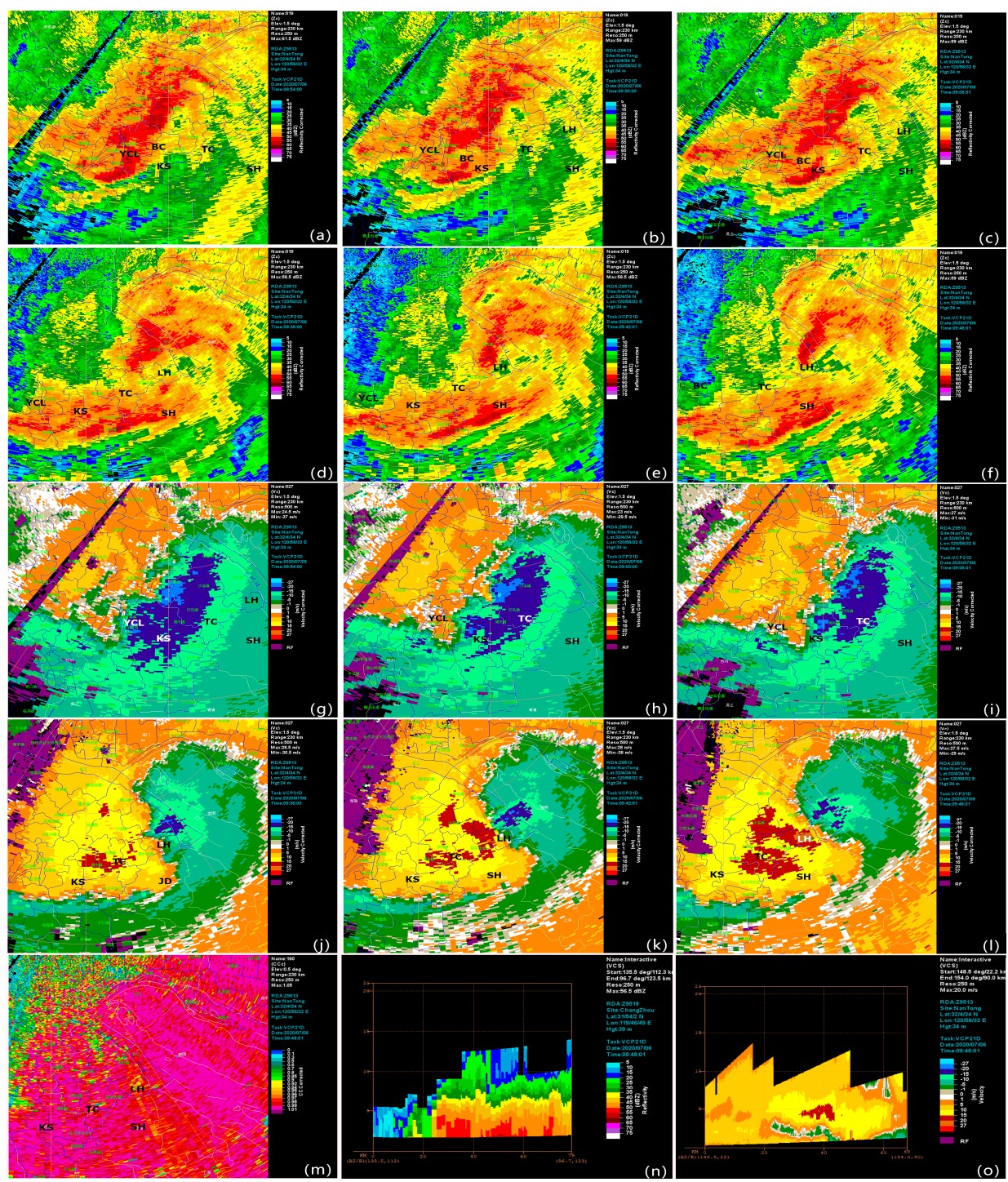

**Figure 6.** *Cont.*

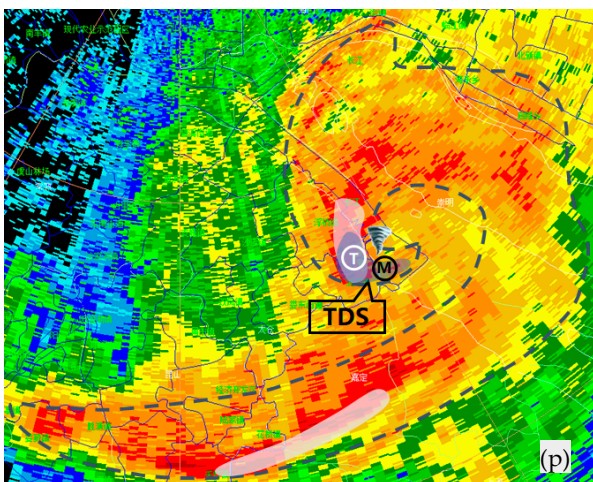

**Figure 6.** The 1.5° elevation basic reflectivity factor at 08:54 BT (**a**), 09:00 BT (**b**), 09:06 BT (**c**), 09:36 BT (**d**), 09:42 BT (**e**), and 09:48 BT (**f**) at the Nantong Radar Station on 6 July 2020. The 1.5° elevation radial velocity at 08:54 BT (**g**), 09:00 BT (**h**), 09:06 BT (**i**), 09:36 BT (**j**), 09:42 BT (**k**), and 09:48 BT (**l**) at the Nantong Radar Station on 6 July 2020. CC at 09:48 BT (**m**). Reflectivity factor profile (**n**). Radial velocity profile (**o**) of Nantong Radar Station at 09:48 BT on 6 July 2020. Schematic diagram of the conceptual model of the Taicang Liuhe tornado (**p**).

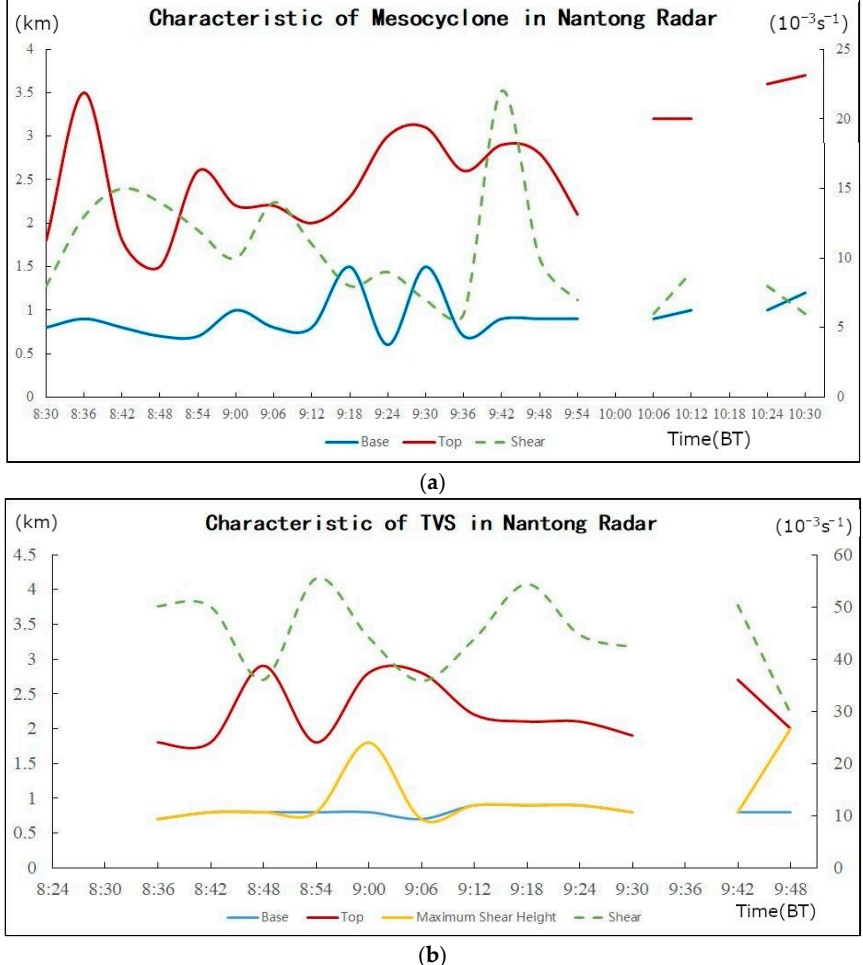

**Figure 7.** Characteristics of the mesocyclone (**a**) and of the TVS (**b**) from 08:30–10:30 BT at the Nantong Radar Station on 6 July 2020.

Table 1 shows the TVS features identified by the Nantong radar at 08:24–09:48 on 6 July 2020, where AVGDV represents the average difference velocity, LLDV represents the lowest-level difference velocity, MXDV represents the maximum difference velocity, depth indicates the thickness of the 3D circulation feature, BASE and TOP indicate the bottom and top heights of the 3D circulation feature, MXSHR indicates the maximum shear variable of the 3D circulation feature, and HGT indicates the height. The lowest-elevation radial difference velocity (LLDV) of the tornadoes on Nantong's TVS at 08:24–09:48 was above 10 ms and above 25 m·s$^{-1}$ most of the time; it reached a maximum of 29 m·s$^{-1}$ at 08:54 pm in Bacheng, Kunshan, and a maximum of 27 m·s$^{-1}$ at 09:42 pm in Liuhe, Taicang. The MXDV appeared to increase gradually before the tornado occurred, and the height of the MXDV also decreased gradually. The MXDV was 28.5 m·s$^{-1}$ at the time the tornado occurred in Bacheng, Kunshan, reaching a minimum of 0.8 km. The maximum velocity difference was 27 m·s$^{-1}$, and the minimum was 0.8 km at 09:42. Therefore, it can be seen that tornadoes occurred at the moment when the LLDV and MXDV were maximal (Table 1).

**Table 1.** TVS parameters of Nantong Radar Station on 6 July 2020.

| Time /BT | Bottom Height/km | Top Height/km | Shear Value/ $10^{-3}$ s$^{-1}$ | Maximum Shear Height/km | AVGDV/ m·s$^{-1}$ | LLDV /m·s$^{-1}$ | MXDV /HGT m·s$^{-1}$/km | Depth/ km |
|---|---|---|---|---|---|---|---|---|
| 8:24 | 1.2 | 4 | 40.8 | 1.2 | 18 | 31 | 30.5/1.2 | 2.8 |
| 8:36 | 0.7 | 1.8 | 50.1 | 0.7 | 16 | 26 | 25.5/0.7 | 1 |
| 8:42 | 0.8 | 1.8 | 50 | 0.8 | 21 | 26 | 25.5/0.8 | 1.1 |
| 8:48 | 0.8 | 2.9 | 36 | 0.8 | 13 | 20 | 19.5/0.8 | 2.1 |
| 8:54 | 0.8 | 1.8 | 55.4 | 0.8 | 13 | 29 | 28.5/0.8 | 1.1 |
| 9:00 | 0.8 | 2.8 | 44.1 | 1.8 | 19 | 24 | 23.5/0.8 | 2.1 |
| 9:06 | 0.7 | 2.8 | 35.8 | 0.7 | 15 | 18 | 19/1.8 | 2.1 |
| 9:12 | 0.9 | 2.2 | 43.8 | 0.9 | 14 | 26 | 26/0.9 | 1.2 |
| 9:18 | 0.9 | 2.1 | 54.3 | 0.9 | 14 | 332 | 32/0.9 | 1.2 |
| 9:24 | 0.9 | 2.1 | 44.6 | 0.9 | 18 | 26 | 26/0.9 | 1.2 |
| 9:30 | 0.8 | 1.9 | 42.3 | 0.8 | 17 | 24 | 23.5/0.8 | 1.1 |
| 9:42 | 0.8 | 2.7 | 50.3 | 0.8 | 22 | 27 | 27/0.8 | 2 |
| 9:48 | 0.8 | 2 | 29.6 | 2 | 17 | 14 | 17/2.0 | 1.2 |

### 3.3. Analysis of Triggering and Formation Mechanisms

Tornadoes usually occur under a certain background of large-scale circulation. In this event, there existed an ultra-low altitude jet which strengthened continuously. The ultra-low altitude jet transported warm and humid air and energy to Suzhou, which maintained the precipitation for a long time and provided a warm and moist environmental condition for the occurrence of the tornado. The occurrence, development and maintenance mechanisms of this event will be discussed below with reference to the boundary-layer environment, frontogenesis and wind profile radar.

### 3.3.1. Circulation Field Analysis

From the night of 5th July to the early morning of 6th July 2020, the southern part of Jiangsu Province was in a divergent region at 200 hPa, with obvious upper-level divergence (Figure 8a). A mid-latitude upper trough moved eastward at 500–700 hPa, and the western North Pacific subtropical high (hereinafter referred to as the "subtropical high") was distributed in a zonal shape. Affected by the eastward movement of the westerly trough, the western ridge point of the subtropical high inclined from 105°E to 115°E, and the subtropical high ridge line slightly inclined southward to 23°N. The southern part of Jiangsu Province was near the 584 dagpm line on the northwestern side of the subtropical high, with continuous southwest warm and humid air transport (Figure 8b). The northward air flow behind the upper trough and the southward air flow on the periphery of the subtropical high converged over the southern part of Jiangsu Province. A southwest jet existed throughout the entire layer at 700–925 hPa, which formed wind direction shear and

strong wind speed convergence obviously with the northwestern air flow from the north of the Yangtze River. The southwest jet was intensified and lifted northward at 850 hPa on the morning of 6 July 2020, and the southwest wind speed in the southern part of Jiangsu Province increased from 16 m·s⁻¹ to 20 m·s⁻¹ (Figure 8c). The upper-level divergence in front of the westerly trough at 500 hPa and the divergence and positive vorticity advection in front of a high trough caused systematic upward movement over Suzhou. The upper-level divergence, the development of the low-level shear line and the intensification of the jet together formed a favorable circulation background for the development of the tornado.

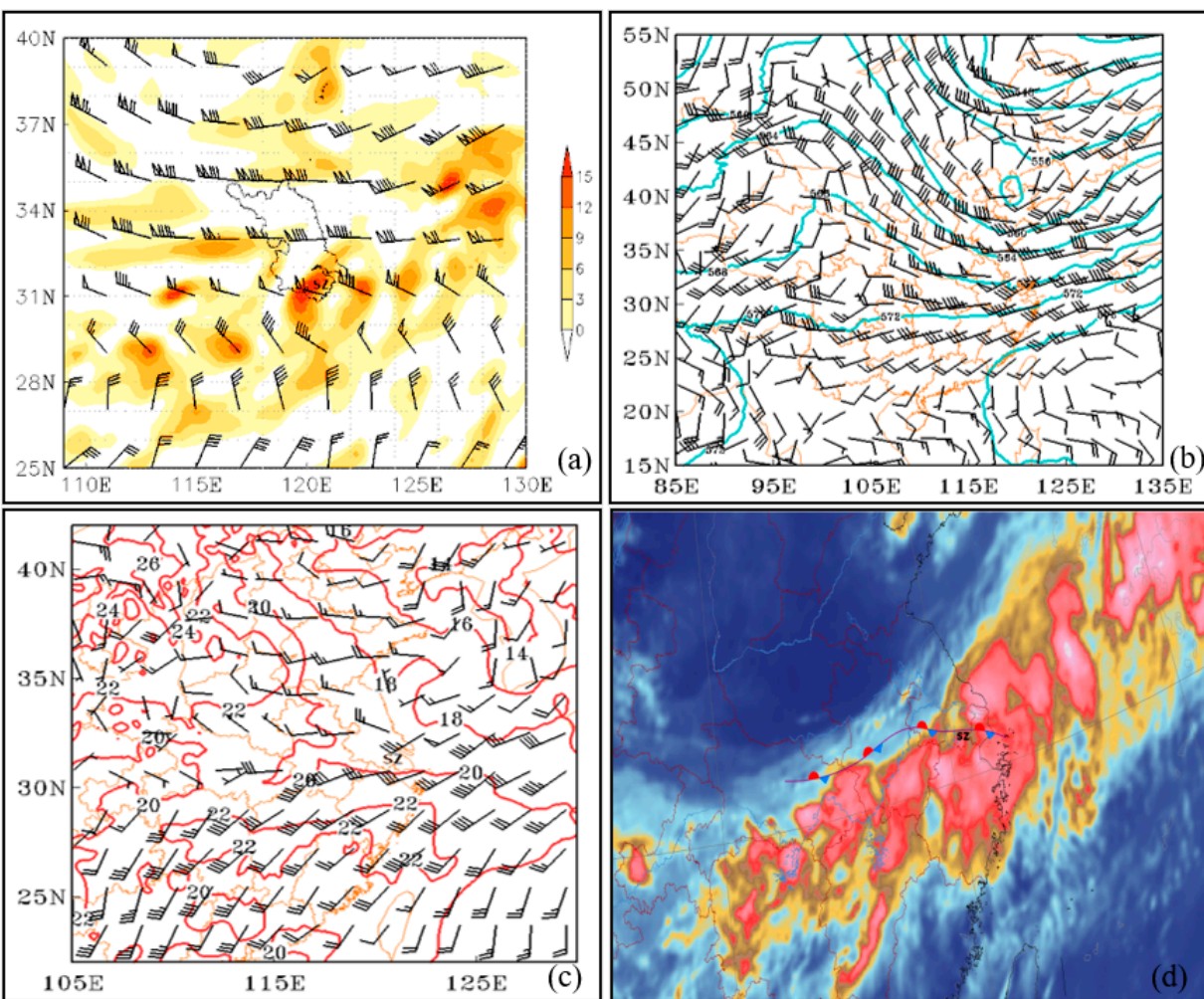

**Figure 8.** Divergence a nd wind field at 200 hPa at 06:00 BT on 6 July 20 20. (**a**) Altitude and wind field at 500 hPa at 06:00 BT (**b**) and at 850 hPa (**c**) at 04:00 BT on 6 July 2020. (**d**) Overlay chart of FY-4A infrared satellite cloud image and ground stationary Mei-Yu front at 08:00 BT on 6 July 2020.

On the ground, cold and warm air converged at the middle and lower reaches of the Yangtze River, forming a quasi-stationary front with a northeast–southwest orientation. Suzhou was on the north side of the eastern section of the quasi-stationary front. The vorticity advection in front of the upper trough, coupled with the wind speed and direction shear on the low level, and the dry air spreading from the Jianghuai region to the middle and lower reaches of the Yangtze River at 700 hPa provided environmental conditions for the development of mesoscale convective systems on the stationary front. In the afternoon of 6 July 2020, the subtropical high and the low-level cold-type shear line moved southward. The mid–low-level southwest jet weakened, and the event tended to its end (Figure 8d).

### 3.3.2. Analysis of Boundary-Layer Environmental Characteristics
Thermodynamic Conditions

The configuration of the synoptic-scale system provided extremely favorable conditions for the torrential rain event accompanied by a tornado. The sounding diagram of Baoshan Station in Shanghai at 20:00 on 5 July showed that there was a low-level jet of 20 m·s$^{-1}$ at 800–700 hPa, with the strongest southwest jet of 24 m·s$^{-1}$ near 700 hPa. The enhancement of the mid–low-level southwest jet increased the wetting and thickening of the moist layer, making the temperature and dew point curves very close together, and the atmosphere was nearly saturated throughout the layer. On the other hand, it exacerbated the instability of atmospheric stratification, with a convective available potential energy (CAPE, the same below) of 939.9 J and a K index of 35.8 °C. The lifting condensation level was near 1000 hPa, the 0 °C level was at 5.6 km, and the cloud-base specific humidity was 20.0 g·kg$^{-1}$, indicating the existence of a warm cloud layer with a high water content and relatively deep layers (Figure 9a). The moderate CAPE value could effectively avoid the reduction in the precipitation efficiency caused by the rapid passage of air blocks with a high water content at low altitude through the bottom of warm clouds. The equilibrium level was 214 hPa, conducive to the development of convection to higher altitudes. The continuous enhancement of the southwest jet and the strong wind direction and speed convergence in the boundary layer created a warm, moist and unstable atmosphere in Suzhou that was favorable to the generation of upward motion. There was strong vertical wind shear in the low level (0–1 km) and the mid level (0–3 km), as the wind was weak near the ground, which was favorable to the organization of the convective system. The low lifting condensation level, the moderate CAPE and the strong vertical wind shear in the mid–low level over Suzhou were all very conducive to the production of heavy rainfall, providing good potential conditions for the occurrence of a tornado.

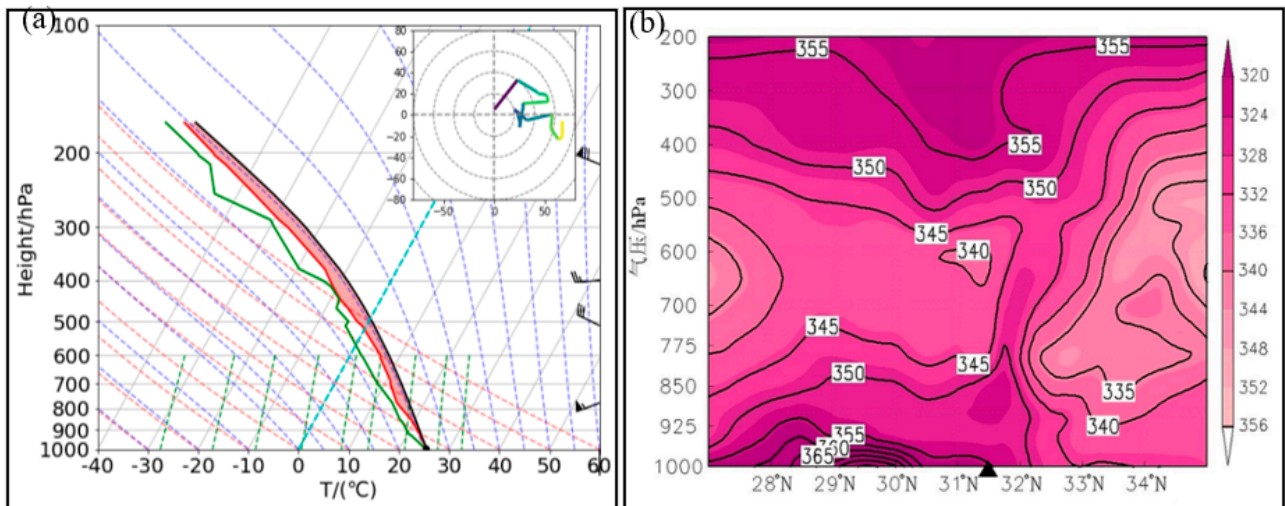

**Figure 9.** (**a**) Sounding map of Baoshan Station at 08:00 BT on 6 July 2020. (**b**) The profile of the pseudo-equivalent potential temperature profile along 120°E (unit: K) at 08:00 on 6 July 2020.

Tornadoes in the Jianghuai region during the Meiyu season tend to occur in a moist and unstable atmospheric environment [12]. A northeast–southwest-oriented high θse band extended to the central and southern part of Jiangsu on the northwest side of the subtropical high, with the θse center value exceeding 365 K. The heavy rainfall occurred near the θse front, where the frontal slope was steep in the low level (below 775 hPa) and tilted with a height above 775 hPa to 500 hPa. Below 600 hPa, there was an unstable layer structure with θse decreasing with height, reflecting the baroclinic characteristics of the atmosphere (Figure 9b).

Dynamic Conditions

Around the morning of 6 July 2020, there was a cold shear line at 850 hPa over southern Jiangsu, and Suzhou was located at the intersection of southwest and southeast airflows at the surface. The near-surface CIN was zero, the cloud base was low, the LCL was close to the surface and the LFC was also relatively low, which was very conducive to the initiation of the initial convection.

The vertical wind field characteristics near the shear line showed (Figure 10a) that there was a clear secondary circulation loop above the north side of the cold shear at 850 hPa, which was a direct thermodynamic circulation loop, and there was a weak indirect thermodynamic circulation loop on the south side of the cold shear. The ascending motion was mainly located between the two circulation loops with opposite directions, and the ascending motion was strongest near the shear line.

Vertical wind shear is a key factor in the development of deep convection [23,24], and strong low-level vertical wind shear provides a source of vorticity for the generation of tornadoes [6–8]. In the rainy season, there was a strong development of cyclonic vorticity before the occurrence of tornadoes, especially in the middle and lower layers, and the strong vertical wind shear in the boundary layer played an important role in the rapid increase in cyclonic vorticity [12]. In this process, the vector difference of the wind shear from 0 to 1 km exceeded 15 m·s$^{-1}$, and the vector difference from 0 to 3 km exceeded 18 m·s$^{-1}$, both reaching the level of strong vertical wind shear in the lower and middle layers, and the environmental conditions were very favorable for the formation of initial vortices (Figure 10b). There was positive relative vorticity below 700 hPa near the shear line and there was a large positive vorticity value center above Suzhou, and the positive vorticity area was a significant negative divergence area, indicating that the wind field convergence was strong (Figure 10c).

The wind profile radar data further confirmed that before the occurrence of heavy precipitation, the whole layer was in a consistent southwest airflow, and the water vapor transport was sufficient. The wind speed increased gradually with a height below 1 km, and the wind speed increased rapidly above 1.5 km, forming a strong low-level jet, with wind speeds exceeding 20 m·s$^{-1}$ from 2 km to 4 km. Then, the low-level jet began to propagate downward. The ultra-low-level jet with wind speeds exceeding 20 m·s$^{-1}$ appeared at 1 km at 08:20. Around 8:48, the wind speed reached 20 m·s$^{-1}$ near 900 m. The southwest jets at different heights and different times formed multiple wind-speed convergence regions of different sizes, causing the environmental wind field to form an unstable stratification due to upper and lower disturbances. Before the occurrence of the Kunshan tornado at 09:00, there was a southeast wind near the ground and a southwest jet above 700 m. The wind direction turned clockwise with height, indicating that the near-surface warm advection increased. The wind speed exceeded 30 m·s$^{-1}$ from 1 km to 2 km, and the vector difference of wind from 0 to 1 km reached 21 m·s$^{-1}$. The vertical wind shear of wind direction and wind speed at the lower layer developed to the strongest level, which was very conducive to the formation and intensification of horizontal vortex tubes. When the mesoscale vortex approached, the horizontal vorticity generated by the mesoscale vortex may also have undergone a significant increase in the ability to convert to vertical vorticity due to the ascending motion in the strong vertical wind shear environment when entering the storm along the storm inflow direction compared with 08:00. The low-level warm advection remained, and the wind field disturbance appeared at 900 m from 09:12 to 09:18. The wind reversed with height from 720 to 900 m, indicating that cold air invaded. At 09:36, the southerly wind below 900 m reversed to the northwest wind, corresponding to the peak of the surface hourly precipitation in Taicang, and then the Liuhe tornado occurred in Taicang. At 9:54, the whole layer turned to the northwest wind, and the precipitation quickly weakened. The process in Suzhou ended (Figure 10e).

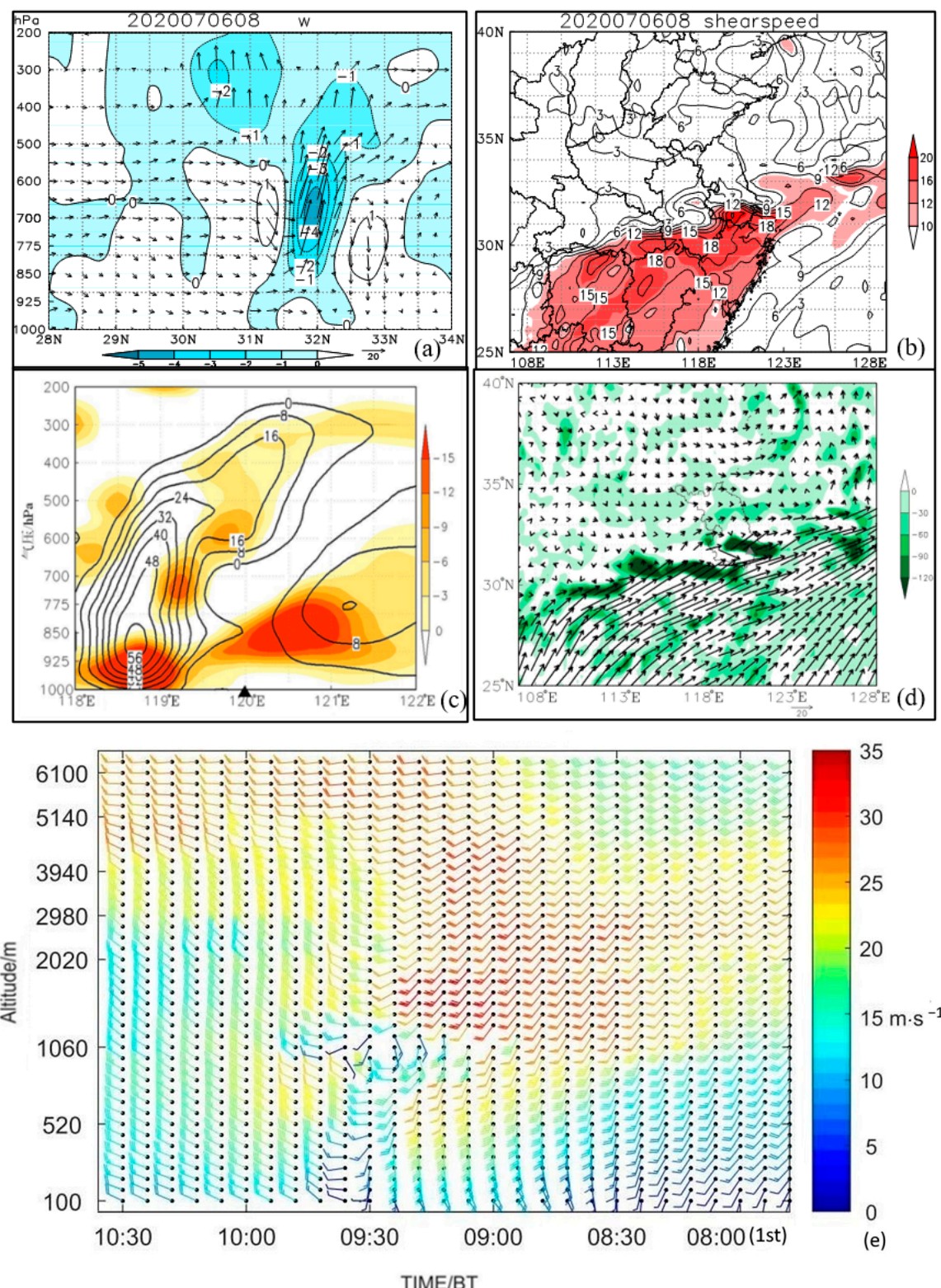

**Figure 10.** Vertical velocity time–height profile along 120°E (**a**). The profile of the pseudo-equivalent potential temperature latitude–altitude profile along 120°E (unit: K) (**b**). The profile of vorticity and divergence along 120°E (**c**). The vapor flux and the water vapor flux divergence at 850 hPa at 08:00 on the 6 July 2020 (**d**). Time–altitude profile of horizontal wind field for wind profiler radar of TaiCang from 08:00–10:36 on 6 July 2020 (**e**).

Surface Convergence Line Analysis

Tornadoes are usually closely related to surface meso- and small-scale convergence. From 05:00 BT on 6 July 2020, a mesoscale low pressure with a center at 31.3°N and a central pressure of 996.3 hPa moved eastward from the south of Nanjing and the border of Anhui. There was a convergence line formed by southerly and easterly winds near and at the front side of the low-pressure center. The main precipitation echo was located in the central and northern part of Suzhou, mainly stratiform precipitation. The low-pressure center with a central pressure of 999 hPa moved northeastward to 31.7°N at 08:00 BT. The southerly wind at the south and southeast side of the low-pressure center intensified, and the wind speed convergence near the low-pressure center was obvious. The low-pressure center continued to move eastward at 09:00 BT, and the southerly wind at the south and southeast side further intensified. A weak northwesterly wind appeared at the northwest side, and the wind direction near the center began to show rotational characteristics. The surface convergence line moved eastward along the line from the north of Kunshan to the south of Taicang. The precipitation echo band along the convergence line had continuous generation and development of mesoscale convective cloud clusters, forming a train effect. The precipitation corresponding to Jiangxiang Village in Kunshan increased rapidly. The Kunshan Bacheng tornado occurred on the right front side of the surface convergence center movement direction. At 10:00, the low-pressure center moved eastward to Zhangjiagang in Suzhou. The wind field near the low-pressure center rotated counterclockwise. The convergence at the front side of the low pressure reached the strongest level, providing cyclonic vorticity conditions for the Taicang Liuhe tornado (Figure 11a–d).

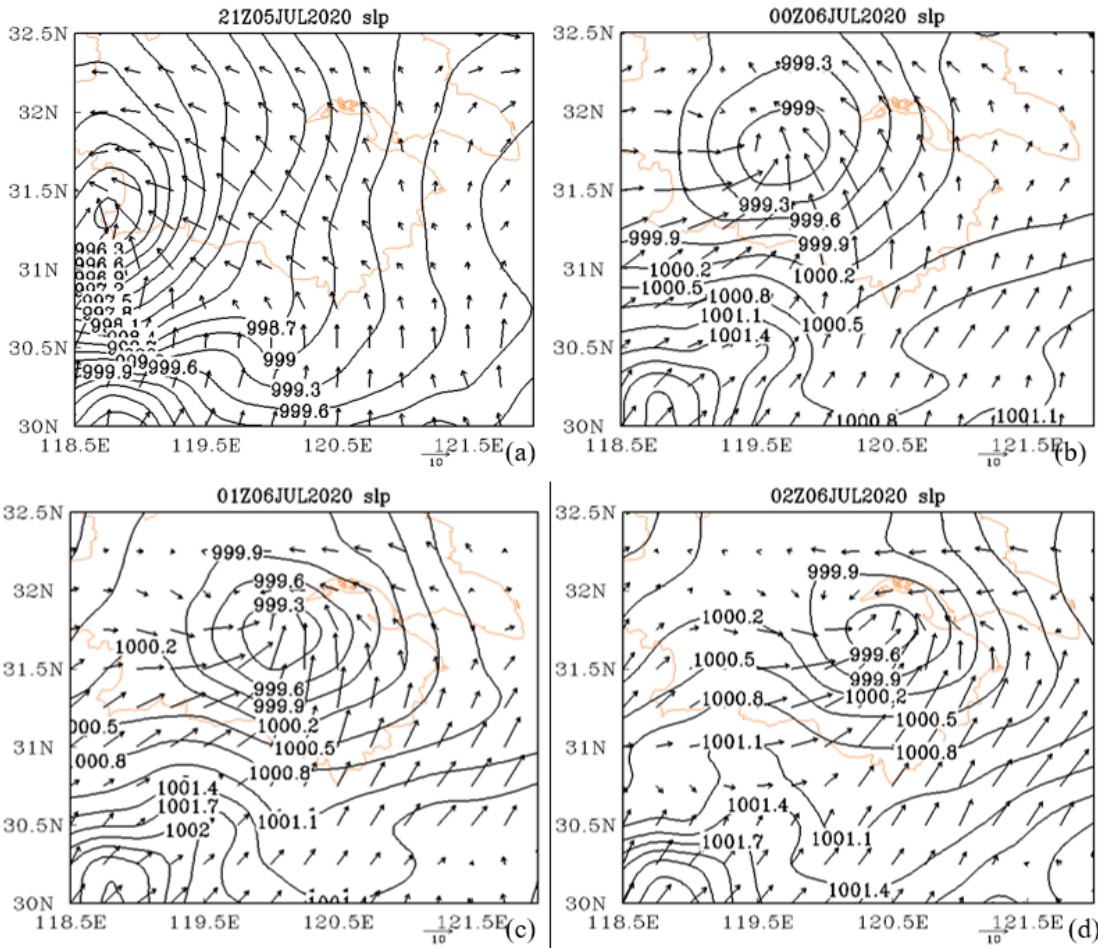

**Figure 11.** *Cont.*

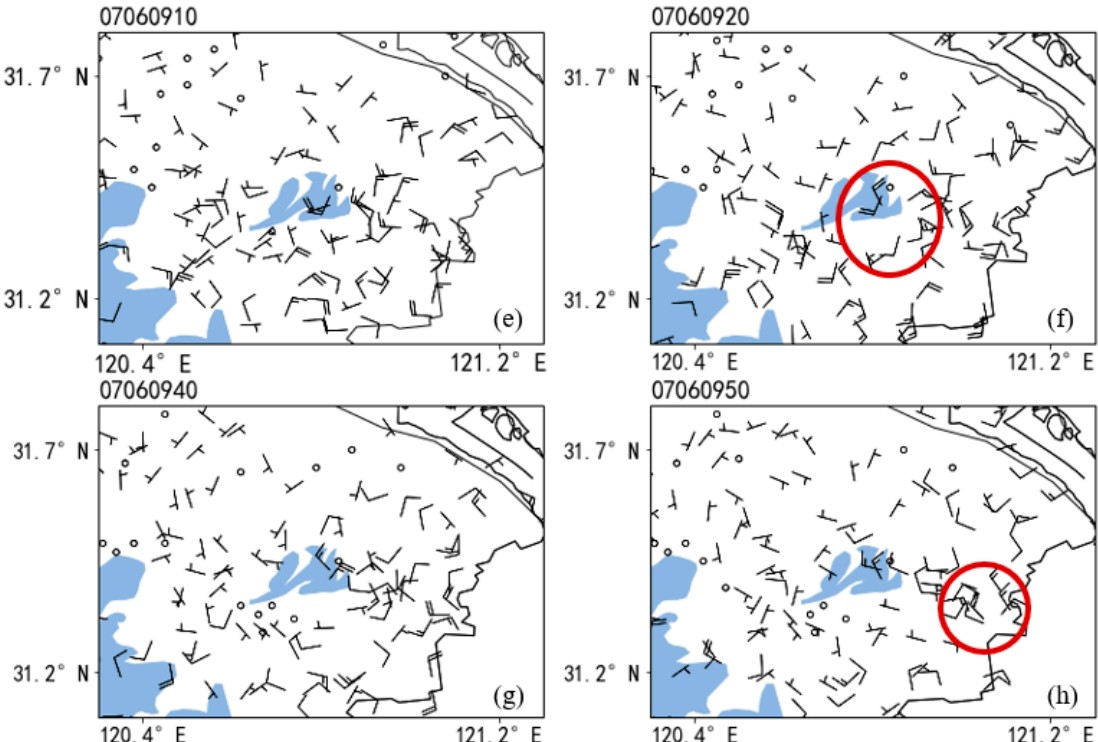

**Figure 11.** Pressure and wind field on the ground at 05:00 BT (**a**), 08:00 BT (**b**), 09:00 BT (**c**), and 10:00 BT (**d**) on 6 June 2020. Ground automatic station wind field of southern Suzhou at 9:00 BT (**e**), 9:20 BT (**f**), 9:40 BT (**g**), and 9:50 BT (**h**) on 6th July 2020 (the blue shaded area is Taihu Lake).

The surface wind field data from the automatic weather stations can more clearly show the evolution of the surface wind field before the tornado's occurrence. There was a small-scale vortex on the east bank of Yangcheng Lake in the southerly wind field at the right front side of the mesoscale low-pressure movement from 09:10 to 09:20, corresponding to the Kunshan Bacheng tornado at 09:10. On the surface wind field of the automatic weather station, there was another small-scale vortex near Taicang Liuhe from 09:40 to 09:50, with the scale smaller than that of the Kunshan Bacheng vortex. The above analysis shows that this tornado event occurred in the warm area far away from the cold and warm air boundary, without an obvious temperature gradient, which was due to the cold air in the lower part of the cold area always being colder and thicker, such that a tornado could not be generated. The small-scale vortex moved along the direction of the surface convergence line, and the tornado vortex developed strongly to produce a tornado. Its movement path was consistent with the movement path of the small-scale vortex. Under the specific environmental background, the surface convergence line and the meso- and small-scale vortex are important systems for the occurrence and development of tornadoes, and the analysis of the surface wind field is helpful for the forecasting and early warning of tornadoes (Figure 11e–h).

Moisture Conditions

There was obvious moisture transport at 850 hPa and 700 hPa in this process, and the moisture mainly came from the Bay of Bengal. At 09:00, there was a strong moisture convergence center along the Jiangsu River and the southeast of Jiangsu, with the center value exceeding $120 \times 10^{-5}$ kg$^{-1}\cdot$hPa$^{-1}\cdot$m$^{-2}\cdot$s$^{-1}$, indicating obvious moisture accumulation, providing sufficient moisture conditions for the occurrence of heavy precipitation (Figure 10d).

We further used the data of the Kunshan microwave radiometer to analyze the temperature and humidity conditions of this process. In terms of the temperature field, from the

early morning of the 6th, the stationary front remained over Suzhou. It was in an unstable stratification, dry and cold above and warm and humid below, from 03:00 to 08:30. The temperature in the high altitude of 8–10 km rose rapidly from 8:30 to 10:00, indicating latent heat release of condensation at the middle and high altitudes with the occurrence of heavy precipitation. There was a saturated area with a relative humidity of more than 90% from 4000 m to the ground during the process, indicating a thick moist layer. The relative humidity above 8000 m was below 30% from 03:00 to 05:00, indicating a dry layer. After 05:00, there was a short-term downward penetration of dry and cold air, corresponding to the first peak of precipitation. From 07:30 to 10:00, the lower boundary of the dry layer was stable at 6000 m. Especially within one hour around 09:00, the liquid water at 3500–4500 m and the water vapor density below 2000 m both increased significantly. Especially from 08:30 to 09:00, there was a strong center of water vapor density exceeding 40 g·cm$^{-1}$ and liquid water exceeding 10 g·cm$^{-1}$, corresponding to the second peak of precipitation, indicating that the microwave radiometer data have some indicative significance for the intensification of the convective system (Figure 12).

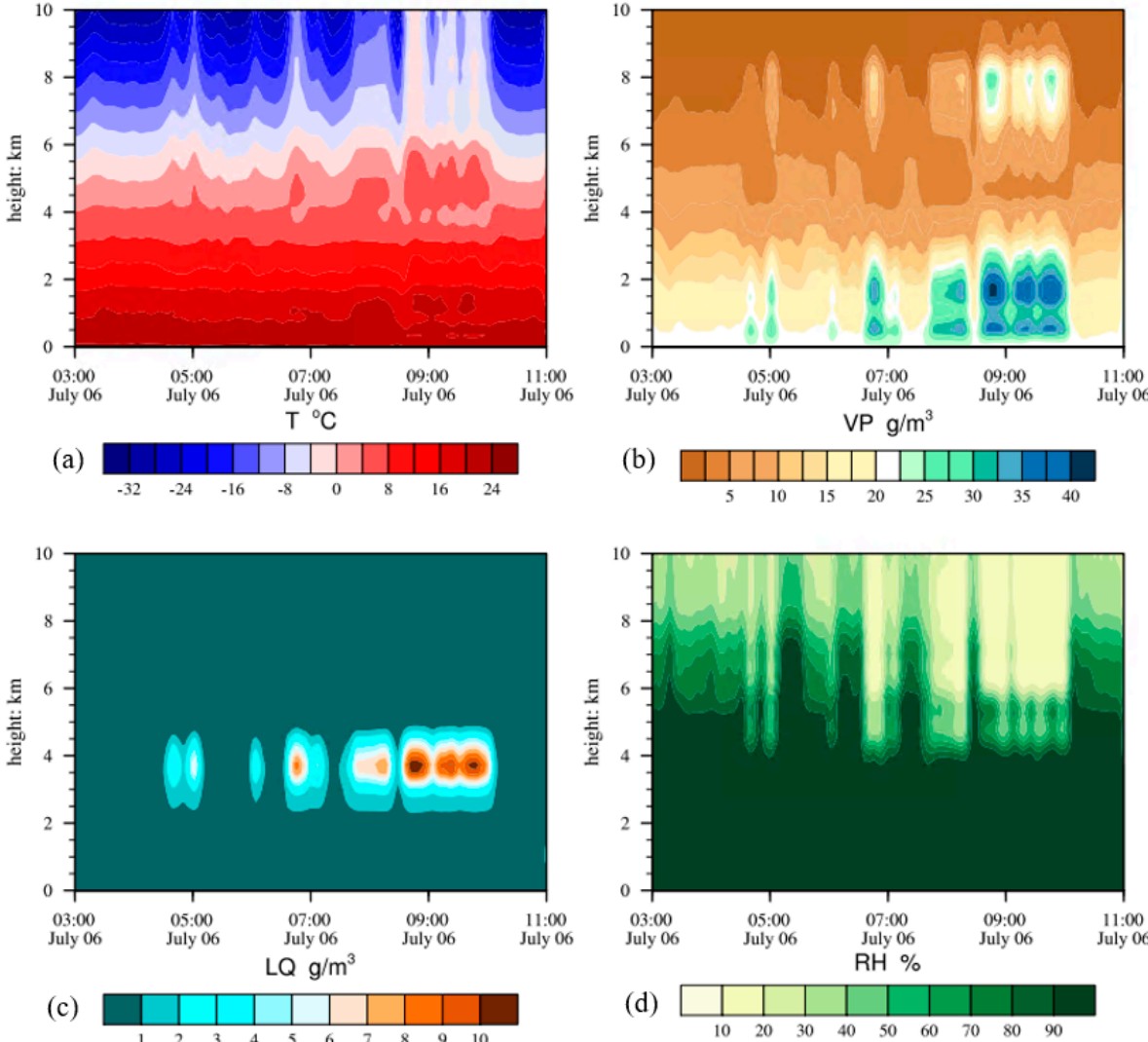

**Figure 12.** Microwave radiometer temperature profile (**a**), moisture density (**b**), liquid water (**c**), and relative humidity profile (**d**) at Kunshan Station at 3:00 to 11:00 am on 6th July 2020.

*3.4. Lightning Frequency*

Strong convective weather is often accompanied by lightning activity, and the characteristics of lightning activity can have a certain indicative effect on deep wet convective

weather [25,26]. The stronger the convective activity, the more obvious the indicative effect. REAP et al. [27] and BRANICK [28] et al. found that lightning occurrence has a certain indicative effect on severe convective weather, and lightning change characteristics can be used as indicator factors of severe convective weather through study of the lightning characteristics of hail, rainstorm and tornado weather processes.

It can be seen from the hourly lightning frequency distribution map from 07:00 to 11:00 BT for 6 July (Figure 13a–d) 2020 within Suzhou that the frequency of lightning was relatively low from 07:00 to 08:00, and sporadic lightning was only detected in Xangcheng and Taicang. The frequency of lightning significantly increased from 08:00 to 09:00, mainly concentrated in Xangcheng and Kunshan, located in central Suzhou, which is consistent with the spatial distribution of heavy precipitation on the Mei-yu Belt, during which the Kunshan BaCheng tornado touched ground at 08:54. As the heavy rainfall belt moved eastward, the high lightning frequency area moved eastward to Kunshan and Taicang from 09:00 to 10:00, obviously weakened compared to the previous hour. Corresponding to the evolution of mesocyclones, it can be seen that the continual strengthening of the low-level mesocyclones was beneficial for the enhancement of updrafts. The strong convection raised the height of the lower charge zone of the storm, which may be one of the reasons for the decrease in flash frequency to the ground. From 10:00 to 11:00, as the convective system moved into Shanghai, the frequency of lightning rapidly decreased within Suzhou.

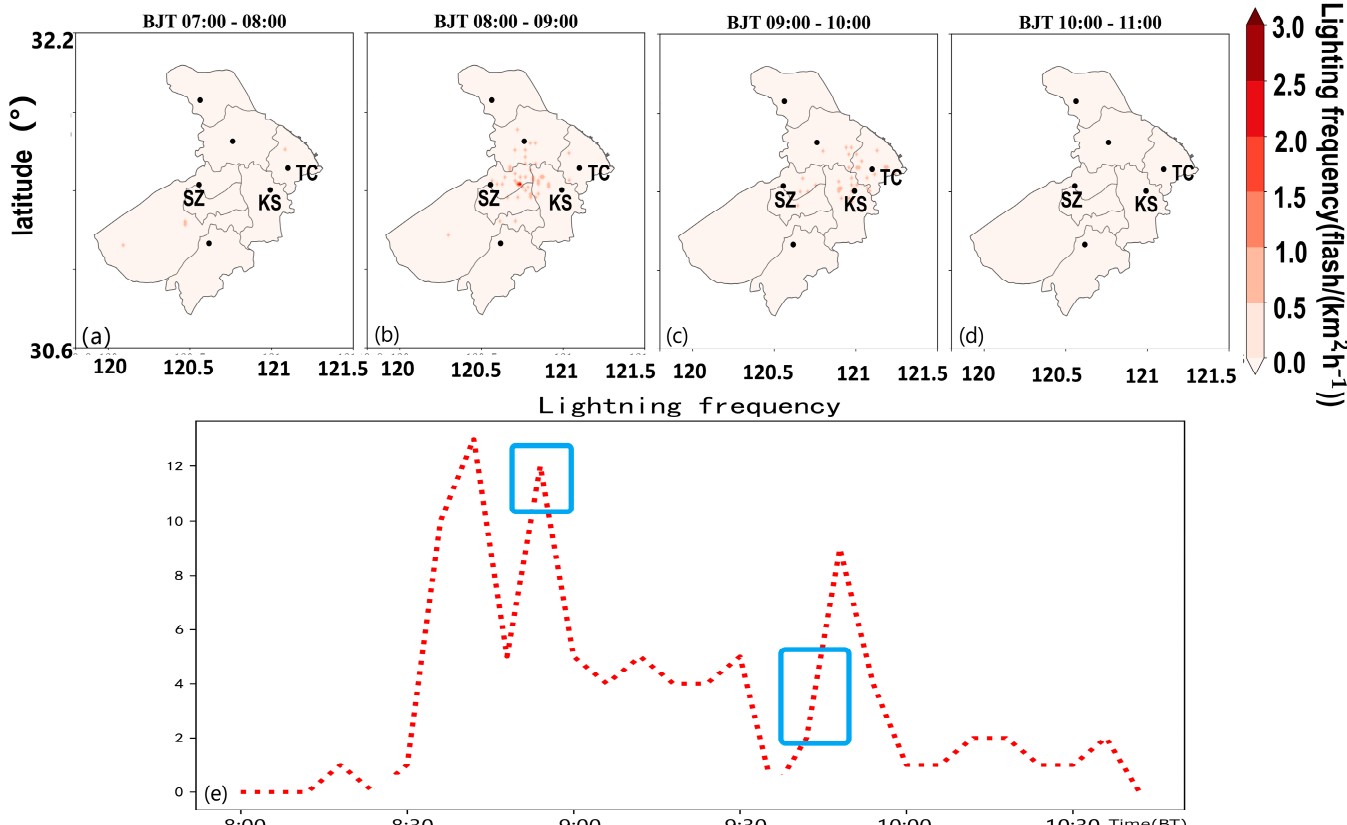

**Figure 13.** Lightning frequency distribution map per hour from 07:00 to 08:00 (**a**), 08:00 to 09:00 (**b**), 09:00 to 10:00 (**c**), and 10:00 to 11:00 (**d**). The variation in lightning frequency per 6 min from 08:00 to 10:40 on 6 July 2020 (the blue rectangle represents the moment when the tornado hit the ground) (**e**). (SZ represents Suzhou, KS represents Kunshan, TC represents Taicang).

A clear feature of the evolution of lightning frequency is that the tornado process experienced three lightning jumps at 08:36, 08:54 and 09:48 (Figure 13e). Before the Bacheng tornado touched the ground at 08:36, the lightning frequency experienced first a jump and quickly reached the first peak when the Bacheng tornado was at the formation stage and

had not touched the ground yet. The first lightning jump appeared 15 min and 66 min earlier than the Kunshan Bacheng tornado and the Taicang Liuhe tornado, respectively. At 08:54, when the Kunshan Bacheng tornado touched the ground, the lightning frequency experienced a second jump and reached a sub-peak. Afterwards, the lightning activity transient weakened. At 09:48, the lightning frequency experienced the third jump and reached its third peak after the Taicang Liuhe tornado touched the ground. The tornado occurred during the stage when the lightning frequency reached its peak and then fell back. The above research indicated that a lightning jump has a certain warning effect with respect to the impact of tornadoes. When there are multiple jumps in lightning frequency, it is necessary to be vigilant, as there may be strong weather, such as hail or tornadoes.

## 4. Discussion

In this paper, we have shown that during the eastward movement of the low vortex, small-scale vortices were triggered and developed, forming two tornadoes successively in the eastern section of the Mei-yu front. Overall, our studies establish that the mesocyclones were detected with decreasing heights and increasing shear strengths and that the low-level vertical wind shear of 0–1 km increased significantly before and after the tornado occurrence, which was more conducive to the formation and intensification of horizontal vorticity tubes. Although there are important discoveries revealed by these studies, there are also limitations. First, under different weather backgrounds, there are many differences in the environmental conditions of tornadoes. Even under similar favorable environmental conditions, the probability of tornado formation is extremely low. Therefore, the key mechanism and factors of tornado formation are not yet fully understood, and the key physical factors that form tornadoes are not yet fully clear. Second, against the background of the Mei-yu, the environmental conditions conducive to tornadoes usually cover a large geographical range, but the number of tornadoes is usually only one or several. Third, microwave radiometers are greatly affected by environmental factors, and measurement results may have some errors. Due to the timing of the lightning location data for Suzhou, the conclusion of this article is only based on a typical tornado storm. With the accumulation of data, the understanding of microwave radiometer and lightning activity characteristics during the tornado process will become clearer. Overall, in the future, with the development of denser ground automatic weather station networks and high-resolution precision radar observations, there will be a deeper understanding of the detailed structural features and tornado eddies of tornado storms.

## 5. Conclusions

This paper used Doppler weather radar, wind profiler radar, ERA5 reanalysis data, ground-based automatic stations and other multi-source data to comprehensively analyze two consecutive tornado events associated with heavy precipitation during the Meiyu season and to detail the formation and development process of the tornadoes and the triggering and maintenance mechanisms of the heavy rainfall. The results show that:

(1) This process was influenced by the quasi-stationary Mei-yu front and the development of mesoscale convective systems at the Mei-yu front, resulting in tornadoes associated with heavy precipitation. The precipitation process had the characteristics of suddenness, extremeness and high precipitation efficiency. During the eastward movement of the low vortex, small-scale vortices were triggered and developed, forming two tornadoes successively in the eastern section of the Mei-yu front.

(2) Heavy rainfall was mainly caused by the eastward movement of heavy rainfall supercell storms. The presence of a gap at the front side indicates that strong incoming airflow entered the updraft. Mesocyclones were detected with decreasing heights and increasing shear strengths. The TVS bottom height dropped to 0.7 km, and the shear value increased to $55.4 \times 10^{-3}$ s$^{-1}$. Tornado debris characteristics (TDSs) could be seen with a low CC value area of 0.85–0.9 in the mesocyclone. The difference between

the lowest-elevation radial velocity (LLDV) and the maximum radial velocity (MXDV) reached the largest value when the tornadoes occurred.

(3) The continuously enhanced low-level jet propagated downward to form a super-low-level jet, and the strong wind direction and wind speed convergence in the boundary layer created a warm, moist and unstable atmosphere in Suzhou. Mesoscale convective cloud clusters were continuously triggered and developed on the east–west convergence line on the ground. With the entrainment of dry air, the southwest dry jet and the southeast moist jet stimulated the formation of a miniature supercell.

(4) The extremely low lifting condensation level and the extremely strong low-level vertical wind shear provided environmental conditions for the formation of the tornadoes. The low-level vertical wind shear of 0–1 km increased significantly upon tornado occurrence, which was more conducive to the formation and intensification of horizontal vorticity tubes. Encountering updrafts and downdrafts, vorticity tubes might have been stretched and intensified. The first lightning jump appeared 15 min and 66 min earlier than the Kunshan Bacheng tornado and the Taicang Liuhe tornado. The Liuhe tornado occurred during the stage when the lightning frequency reached its peak and then fell back.

**Author Contributions:** Conceptualization, methodology and investigation, S.C. and G.H.; resources, data curation and validation, Y.W. (Yi Wang) and P.S.; writing—review and editing, Y.H. and Y.W. (Yue Wu). All authors have read and agreed to the published version of the manuscript.

**Funding:** This research was supported by the "The Pearl River Talent Recruitment Program of Guangdong" (2019ZT08G669); the Youth Innovation Team of China Meteorological Administration (CMA2023QN06); the Jiangsu Province Key R&D Plan Social Development General Project (BE2022851); the China Meteorological Administration's "Unveiled and Commanded" Project (CMA-JBGS202212); and the Basic Research Fund of CAMS (2021Z003).

**Data Availability Statement:** Some or all data, models, or code generated or used during the study are available in a repository or online in accordance with funder data retention policies (Provide full citations that include URLs or DOIs.)

**Conflicts of Interest:** The authors declare no conflict of interest.

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
