# Peer review of "Radar Characteristics and Causal Analysis of Two Consecutive Tornado Events Associated with Heavy Precipitation during the Mei-Yu Season"

_remotesensing, doi:10.3390/rs15235470_

Round 1
Reviewer 1 Report
Comments and Suggestions for Authors
Review for Article no 2642611 entitled:
Radar Characteristics and Causal Analysis of Two Consecutive Tornadoes Event Associated with Heavy Precipitation during the Meiyu Season
by
Shuya Cao, Guangxin He, Yi Wang, Peifeng Shen, He Yan, Yue Wu
Comments
l The article presents a throughout analysis of a variety of significant data including weather stations, radar and mesoscale data conducive to formation and intensification. The results offer a useful contribution to tornadogenesis and intensification mechanisms and in this frame the paper can be accepted for publication after considering some minor suggestions in the following.
l In Abstract, line 18 “the Meiyu Season” should be defined in about a couple of lines.
l In Abstract, line 22, explain the term “gap”, i.e. radar reflectivity, etc.
l In Abstract, all the acronyms should be given in full words in the first time appeared in the text, i.e. Tornado Vortex Signature (TVS), Tornado Debris signature (TDS), etc.
l In 1.Introduction the line 55 has something missing that should be completed.
l In 1.Introduction, lines 68 to 73 is a too long paragraph that should be shortened or divided in two so giving a more clear meaning.
l In 2.Materials and Methods, in Figure 1, locations with names of a few major cities should be displayed in the image.
l In 3.Results in Figure 2 locations with names of a few major cities should be displayed in the image.
l In Figures 4, 5, 6, 8 the tornado locations should be indicated with an appropriate symbol, circle or mark in the images.
l In Figure 7, axis units should be placed in the graphs, in both x and y axes.
l In Table 1, units for parameters AVGDV, etc. should be added.
Comments on the Quality of English Language.
Author Response
Dear reviewer,
Thank you very much for your suggestions. I have carefully revised each one according to the suggestions, and now I will explain my modifications as follows:
1. I add definition of “the Mei-yu Season” In abstract, line 17-18;
2. I add explain of “gap” In abstract, line 23;
3. I add full words in the first time appeared, line 25-28;
4. Introduction the line 55 Modified;
5. lines 68 to 73 modified,and the full text has been modified;
6. I add locations with names of meteorological station,Doppler radar station,wind profile radar station and microwave radiometer stations in Figure 1;
7. I add locations with names of major cities in Figure 2
8. I add tornado locations In Figures 4, 5, 6, 8;
9. modified,add axis units in Figure 7;
10.modified,add units for parameters AVGDV, etc in Table 1

Reviewer 2 Report
Comments and Suggestions for Authors
In this paper, the occurrence and development of convective systems, the evolution characteristics of mesocyclones and TVS, etc. are analyzed based on multi-source observation data and reanalysis data. The results of this paper look good. The objective of this study falls within the scope of this journal. However, there are still some questions should be strengthened.
(1) It is recommended to rewrite some expressions, e.g., Lines 16-20.
(2) Line 24: “TVS bottom height dropped to 0.7 km, the shear value increased to 55.4 × 10-3 s-1.” There is grammatical error in this sentence.
(3) Lines 54-54: Please check this sentence.
(4) It is recommended to add some related studies in the part of Introduction.
(5) Lines 88-90: The website of ERA5 is recommended to be listed in this paper.
(6) Figure 2: Please change the text to English in the left figure.
(7) Figure 7: Please explain the means of x and y in these two figures.
(8) Lines 354, 362: “s-1”, “kg-1”, please revise these similar errors throughout the paper.
(9) The order of Conclusions and Discussion should be reversed.
(10) The Discussion is recommended to be strengthened.
(11) Some recent references should be added and cited in this paper.
(12) The manuscript should be proofread by an English speaker. There are awkward sentences, word choices and grammar mistakes throughout the manuscript.
Comments on the Quality of English LanguageThe manuscript should be proofread by an English speaker. There are awkward sentences, word choices and grammar mistakes throughout the manuscript.
Author Response
Dear reviewer,
Thank you very much for your suggestions. I have carefully revised each one according to the suggestions, and now I will explain my modifications as follows:
(1) we rewrite some expressions in lines 16-20 and the full paper
(2-3)we rewrite these sentences
(4) we add some related studies in Introduction
(5) we add the website of ERA5,Lines 95
(6) we modified the text to English in Figure 2
(7)we add the explain of x and y coordinate in Figure 7
(8) we modified these errors throughout the paper
(9) we reversed the order of Conclusions and Discussion.
(10) we add some discussion
(11) we add and cited 5 recent references in this pape
(12) we revise the English expression throughout the paper

Reviewer 3 Report
Comments and Suggestions for Authors
Journal: Remote Sensing
Article Radar Characteristics and Causal 1 Analysis of Two 2 Consecutive Tornadoes Event Associated with 3 Heavy Precipitation during the Meiyu Season
Considerations: The subject of the paper is very interesting and important because those phenomena occur in many parts of the world and affect a large number of people. So understanding the details about those occurrences is fundamental to developing better nowcasting tools. The events are well described in different physical and meteorological aspects, since, “Initiation stage of convective storm”, “Development stage of the convective storm”, “Mature stage of the supercell”, analysis of local circulation, as well as thermodynamic and dynamic conditions. This is good because it covers the different aspects of the events. However, there are factors that are also important and can improve the work. For the item “Materials and Methods”, the description is very summarized. The authors just listed the datasets and a few details, for example, spatial resolution for ERA 5 and time of radar images. But didn’t describe which fields going to be analyzed or which radar variables will be studied and why. This is very important because the dual polarization radar, for example, could have many variables and each one has self-characteristics, depending on the aims or what are you looking for in analyses. So, I suggest that the authors improve this part of the paper. Some Figures can be better described too. Starting with Figure 1, that can be added the location of the radars. In Figure 7, the axes needs to be described; it helps the understanding of readers. In Figure 9, there is no description of the sounding diagram, like the time of sounding. Okay, the time of sounding is described in the text, but remember, it should be clear in the figures too. How far is Baoshan station from radars and from the occurrence of tornadoes? Another important observation is in relation to acronyms, some of them were mentioned without transcription, when did it for the first time, such as TVS and LLDV. In order to improve the results, I would like to make some suggestions. The ERA 5 used in this work is a very good reanalysis and this spatial resolution can represent the mesoscale processes better than the others. So, I believe that could be interesting to plot artificial soundings during the stages of supercells. Obviously, the artificial soundings need to be validated with the observed ones. The authors have all the data and evolution of the system and show that. Then, it would be very useful to calculate the track of tornadoes. In fact, I was waiting to see the trajectories made by the tornadoes. This can suggest or indicate important thermodynamics and dynamics characteristics. The lightning analysis has been treated as necessary for nowcasting monitoring. Furthermore, it is possible to use in real-time the Geostationary Lightning Mapper (GLM). One of the reasons is the “lightning jump” preceding the occurrence of severe weather (large hail, wind gusts associated with thunderstorms, and/or tornadoes). In your case, would the lightning jump have preceded the tornado's occurrence? How long? Finally, based on the found results it is possible to develop a schematic graphic of supercell thunderstorms with the locations of features at the different levels, showing for example, TDR, ZDR ring, KDP column, the mesocyclone, and TVS. In general, the results are consistent with the studied events and contribute to the understanding of the evolution of tornados in China. October, 2023.
Author Response
Dear reviewer,
Thank you very much for your suggestions. I have carefully revised each one according to the suggestions, and now I will explain my modifications as follows:
1. I add the trajectories made by the tornadoes in FIG.3c
2. I add the research on GLM, and found 3 “lightning jump” in this tornado event, preceding the occurrence of severe weather
3. I add a schematic graphic of supercell thunderstorms in FIG. 6p, and relevant descriptions have been added in the article.

Round 2
Reviewer 2 Report
Comments and Suggestions for Authors
The authors have addressed all of my comments.